# NeuroFusion: A Unified Framework for Generalized Visual Stimulus Decoding from fMRI Across Datasets and Subjects

**Muhammad Kashif**
Department of Biomedicine and Prevention
University of Rome Tor Vergata
Rome, Italy
muhammad.kashif@uniroma2.it

**Matteo Ferrante**
Department of Biomedicine and Prevention
University of Rome Tor Vergata
Rome, Italy
matteo.ferrante@uniroma2.it

**Nicola Toschi**
Department of Biomedicine and Prevention
University of Rome Tor Vergata
Rome, Italy
Martinos Center for Biomedical Imaging, MGH
and Harvard Medical School, Boston, USA
nicola.toschi@uniroma2.eu

## Abstract

Recent advancements in neural decoding have shown promising results in reconstructing visual experiences from brain activity. However, existing approaches focus primarily on decoding within a single dataset or subject, which limits generalization across various sources of neuroimaging. In this work, we propose a novel framework for the decoding of visual stimuli **between subjects and between data sets**, integrating neural recordings from multiple publicly available fMRI datasets. To address inherent intersubject and interdataset variability, we introduce a contrastive learning-based alignment strategy using image embeddings from a pre-trained IP-Adapter model. Our approach learns a shared latent space by aligning subject-specific neural representations with image features, enabling generalized decoding across both subjects and datasets. In addition, we propose a simple yet effective data augmentation method using ridge regression. This method synthesizes realistic fMRI-like signals from novel images by predicting voxel activity and injecting learned noise distributions, thus enhancing training diversity and model robustness. To the best of our knowledge, while several recent studies have explored cross-subject decoding, our work is the first to extend this direction to joint decoding across multiple datasets and subjects using a unified training framework. Empirical results show that our method achieves competitive and, in some metrics, state-of-the-art decoding performance in this more challenging and realistic setting.

## 1 Introduction

Human perception integrates diverse sensory inputs such as vision, audition, and touch through complex neural computations [1, 2]. Among these, visual stimuli are particularly central, yet the mapping of stimuli to brain activity is indirect and shaped by cognition, attention, and individual variability [3, 4, 5]. Brain decoding aims to reconstruct external stimuli from neural activity, providing insight into perception and mental representation [6, 7, 8]. Functional magnetic resonance imaging

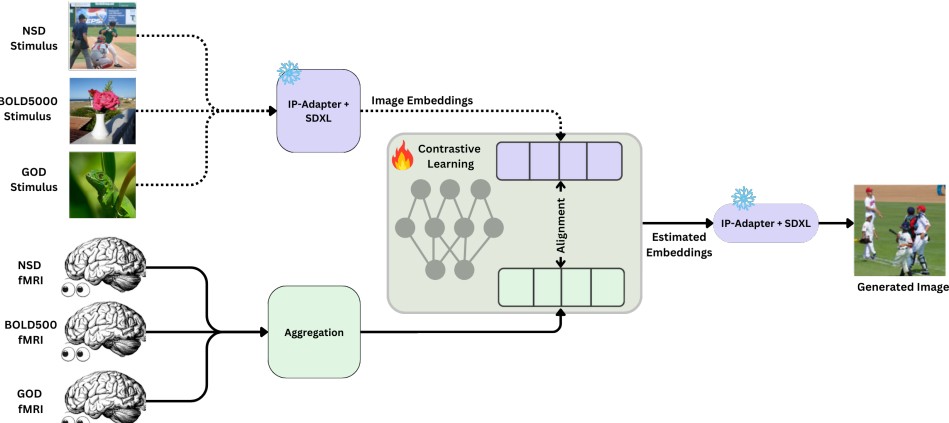

Figure 1: Overview of our framework. Visual stimuli from NSD, BOLD5000, and GOD are embedded via IP-Adapter + SDXL. Corresponding fMRI signals are aligned to these embeddings using contrastive learning, enabling reconstruction via the same generative pipeline.

(fMRI) is widely adopted due to its high spatial resolution and whole brain coverage [9], although it suffers from low temporal resolution [10, 11]. Recent advances in deep learning, such as GANs and diffusion models, have enabled increasingly realistic and semantically faithful reconstructions from fMRI data [12, 13, 14, 15]. These models typically project brain activity into image embedding spaces to enable perceptual decoding [16, 17, 18]. Most prior work has focused on subject-specific decoding, which restricts scalability [19, 20, 21]. More recent cross-subject models aim to generalize between individuals [22, 23, 24, 25, 26], but remain limited to single datasets with uniform acquisition settings (see Appendix B for a detailed literature review). In contrast, practical applications demand decoders that generalize across both subjects and datasets, which introduces challenges such as domain shifts and heterogeneous voxel distributions.

In this work, we present a unified decoding framework that generalizes across subjects and datasets. We integrate multiple fMRI datasets with diverse acquisition parameters and subject pools to examine whether there are shared semantic representations of visual perception between individuals. To this end, we employ vision-language embeddings from Stable Diffusion XL (SDXL) [27] and IP-Adapter [28] to project fMRI signals into a shared semantic space. IP-Adapter injects image embeddings into the cross-attention layers of SDXL, allowing flexible conditioning without full model retraining. Furthermore, we propose a simple but effective data augmentation strategy that synthesizes fMRI-like signals from novel images using ridge regression and noise sampling. This improves training diversity and robustness, especially in low-data or cross-domain settings. Altogether, our framework advances scalable and generalizable brain decoding by combining cross-domain alignment with principled data augmentation, offering a step toward universal neural decoding systems for real-world applications.

## 2 Methods

Our goal is to build a generalizable brain decoding model capable of reconstructing visual stimuli from fMRI signals across diverse subjects and datasets. We propose a modular framework that combines contrastive representation learning, vision-language embeddings, and generative image synthesis. An overview is shown in Figure 1.

Our model projects fMRI signals into the semantic embedding space of a vision-language model. Specifically, we use the IP-Adapter encoder [28] for Stable Diffusion XL (SDXL) [27], built on OpenCLIP ViT-bigG-14 [29], which maps images to global [CLS]-based embeddings. Given fMRI input $x_{\text{fMRI}}$, a neural encoder $f_n(\cdot)$ predicts latent representation $z_{\text{fMRI}}$, which is aligned with the corresponding image embedding $z_{\text{IP}} = h(I)$ using a contrastive loss (Section 2.1). We evaluate our model on three widely used fMRI datasets: NSD [30], BOLD5000 [31], and GOD [32] which differ in scanner type, resolution, subject count, and stimulus distribution. Full dataset details are provided in Appendix C.

## 2.1 Cross-subject and cross-dataset neural vision alignment

We train a contrastive model to align brain responses across subjects and datasets with image embeddings from the IP-Adapter. Let $h(\cdot)$ map image $I$ to embedding $z_{\text{IP}} \in \mathbb{R}^d$, and $f_n(\cdot)$ map fMRI $x_{\text{fMRI}} \in \mathbb{R}^V$ to $z_{\text{fMRI}} \in \mathbb{R}^d$. To account for inter-subject differences, we decompose $f_n(\cdot)$ as:

$$f_n(x_{\text{fMRI}}, s) = g_n(a_n(x_{\text{fMRI}}, s)) \tag{1}$$

where $a_n(\cdot, s)$ is a subject-specific projection module, and $g_n(\cdot)$ is a shared decoder into the semantic space. Embeddings are normalized to the unit hypersphere: $\tilde{z} = z / \|z\|_2$, and similarities are computed via scaled dot product: $\text{logits} = \frac{\tilde{z}_{\text{fMRI}} \cdot \tilde{z}_{\text{IP}}^\top}{\tau}$ We optimize a symmetric contrastive loss: $\mathcal{L} = \frac{1}{2} \left( \mathcal{L}_{\text{CE}}(\text{logits}, \mathbf{t}) + \mathcal{L}_{\text{CE}}(\text{logits}^\top, \mathbf{t}) \right)$ where $\mathbf{t}$ encodes correct fMRI-image pairings. This encourages semantic alignment across diverse subjects and datasets.

## 2.2 Data augmentation

To address limited fMRI data, we propose subject-specific augmentation by synthesizing brain responses for unseen images. For subject $s$, we fit a ridge regression model from image embeddings $\mathbf{Z}_{\text{IP}}^{(s)} \in \mathbb{R}^{N \times d}$ to fMRI signals $\mathbf{X}_{\text{fMRI}}^{(s)} \in \mathbb{R}^{N \times V}$:

$$\mathbf{W}^{(s)} = \arg \min_{\mathbf{W}} \|\mathbf{X}_{\text{fMRI}}^{(s)} - \mathbf{Z}_{\text{IP}}^{(s)} \mathbf{W}\|_2^2 + \lambda \|\mathbf{W}\|_2^2 \tag{2}$$

This model defines $k_s(z_{\text{IP}}) = z_{\text{IP}} \mathbf{W}^{(s)}$, which predicts synthetic fMRI from novel images (sampled from ImageNet). To simulate trial variability, we add noise $\epsilon \sim \mathcal{H}(e)$, where $\mathcal{H}(e)$ is fitted to residuals $e = (\mathbf{X}_{\text{fMRI}} - k_s(\mathbf{Z}_{\text{IP}}))^2$: $\tilde{x}_{\text{fMRI}} = k_s(z_{\text{IP}}) + \sigma \cdot \epsilon$ We generated augmented datasets with varying percentages and noise scales. This improves robustness and model generalization across domains.

## 2.3 Visual reconstruction via ridge-refined embeddings and IP-Adapter-SDXL

We refine the predicted latent $z_{\text{fMRI}}$ via a subject-agnostic ridge regression to better match the IP-Adapter's space: $W_R = \arg \min_W \|Z_{\text{IP}} - Z_{\text{fMRI}} W\|_2^2 + \lambda \|W\|_2^2, \quad z_{\text{reg}} = z_{\text{fMRI}} W_R$ The refined $z_{\text{reg}}$ is injected into the IP-Adapter conditioning stream for SDXL, guiding the generation of coherent visual reconstructions. This two-stage decoding ensures semantic consistency while leveraging the expressiveness of SDXL.

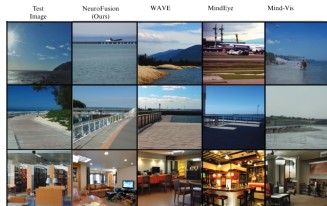 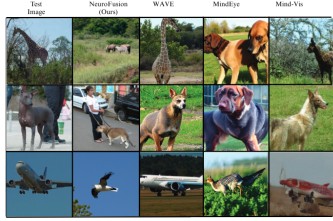 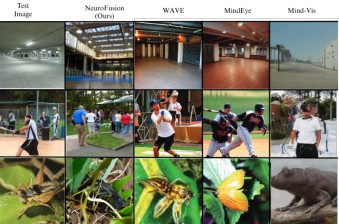

Figure 2: Qualitative reconstructions on BOLD5000. Test images (left) and predicted outputs show preservation of both structure and high-level content.

## 3 Results

We evaluate our framework across NSD, BOLD5000, and GOD datasets under both cross-subject and cross-dataset settings. Decoding performance is assessed using low-level structural metrics (PixCorr, SSIM) and high-level semantic metrics (feature similarities from AlexNet, Inception, CLIP), along with category-level retrieval scores using EfficientNet-B and SwAV. Full quantitative results for BOLD5000 are shown in Table 1; results for NSD and GOD are provided in Appendix D (Tables 2

and 3). Reconstruction examples for BOLD5000 are shown in Figure 2; reconstructed examples for NSD and GOD are shown in appendix D Figures 3 and 5, respectively.

Across datasets, models trained with data augmentation (AUG) consistently outperform their non-augmented counterparts on both structural and semantic metrics. For instance, the "NSD (AUG)" model improves CLIP similarity from 93.0% to 94.8% and SSIM from 0.328 to 0.346 (Appendix Table 2). Similarly, "BOLD5000 (AUG)" raises Inception similarity to 69.2% and Top-1 retrieval accuracy to 19.1% (Table 1).

Compared to prior baselines such as MindReader [23], Brain-Diffuser [21], and MindEye [14], our models achieve competitive or superior results, especially in CLIP and Inception similarity, even without dataset-specific tuning. Notably, the "Cross-model-BOLD5000" and its finetuned variant deliver strong cross-dataset generalization, with SSIM reaching 0.403, matching the best reported values. Results on the GOD dataset (Appendix Table 3) show similarly robust semantic decoding, despite greater domain shift. Our "Cross-model-GOD" variant achieves 81.0% CLIP similarity without finetuning, confirming the ability of shared representations to transfer across acquisition protocols and subject pools. Qualitative results (Figures 3, 2) illustrate that reconstructions preserve high-level perceptual features across datasets. Additional augmentation analysis (Appendix Figure 4) and finetuning evaluation (Appendix Figure 6) confirm that our framework adapts efficiently under limited data and benefits from synthetic neural data augmentation.

Table 1: Image reconstruction on BOLD5000 (structural and semantic metrics). Baselines above the line; our models below. (AUG) = data augmentation.

| Method | PixCorr ↑ | SSIM ↑ | AlexNet(2) ↑ | AlexNet(5) ↑ | Inception ↑ | CLIP ↑ | EffNet-B ↓ | SwAV ↓ | Top-1 acc ↑ |
|---|---|---|---|---|---|---|---|---|---|
| WAVE [24] | **.050** | 0.194 | **69.47%** | **78.31%** | **68.41%** | **78.41%** | .902 | .591 | **20.75%** |
| Mind-Vis [25] | .036 | **.272** | 60.56% | 68.92% | 64.25% | 64.47% | .938 | .593 | 9.19% |
| BOLD5000 | .036 | .374 | 58.9% | 64.0% | 60.2% | 62.2% | .947 | .652 | 10.2% |
| BOLD5000 (AUG) | .033 | .403 | 65.5% | **78.4%** | **69.2%** | 75.7% | **.900** | **.549** | 19.1% |
| Cross-model-BOLD5000 | .037 | .397 | 61.5% | 71.0% | 64.4% | 71.1% | .924 | .580 | 13.4% |
| Cross-model-BOLD5000 (AUG) | .034 | .375 | 68.5% | **79.0%** | **69.9%** | 75.4% | **.911** | **.557** | 19.6% |
| Cross-model-finetuned-BOLD5000 | .040 | .403 | 65.9% | 76.2% | 69.5% | 76.0% | .904 | .553 | 19.3% |

# 4 Discussion

This study demonstrates the feasibility of brain decoding across subjects and across datasets using a unified semantic alignment framework. Unlike conventional models trained per subject or dataset, our method generalizes to varied experimental conditions using contrastive learning with subject-specific alignment layers. Trained on one of the largest brain decoding datasets to date, more than 60,000 images, 13 subjects, and 75 hours of fMRI, the model learns shared visual representations that remain robust across distribution shifts in NSD, BOLD5000, and GOD. This is achieved by aligning fMRI signals with high-level semantic embeddings from IP-Adapter and SDXL [28, 27], allowing strong performance in perceptual and conceptual metrics, even if the fidelity at the pixel level is lower than models such as MindEye [14] or Dream [22].

We show that fine-tuning only the subject-specific alignment layers of a pre-trained model yields strong results, especially in low data regimes, confirming the effectiveness of partial transfer learning, as also seen in recent work [33, 34, 35]. To enhance generalization, we propose a ridge regression-based data augmentation strategy that generates biologically plausible fMRI signals with residual noise. Although this increase improves performance across datasets, its impact is limited by the linear model's ability to capture realistic neural variability -consistent with previous observations [36].

Our findings support growing evidence that contrastive frameworks grounded in foundation models can generalize across subjects and acquisition domains [37, 22]. Mapping diverse brain responses into a shared semantic space reduces individual variance and enables cross-domain transfer, suggesting that the human visual system shares a latent structure learnable via joint training with large-scale visual embeddings. Despite these advances, challenges remain: our dataset still includes only 13 subjects; the augmentation method relies on linear assumptions; and the model prioritizes semantic over structural fidelity. Addressing these may require richer generative priors, improved cortical alignment, or multimodal supervision.

# 5    Conclusion

We proposed a cross-dataset brain decoding model that aligns fMRI signals with vision-language embeddings through contrastive learning and data augmentation. The experiments show a strong generalization across subjects and domains, with minimal fine-tuning. This framework sets the stage for universal BCIs and neural foundation models in real-world applications.

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

## A    Supplementary Material

## B    Related work

Perceptual experiences are intricately encoded in distributed patterns of brain activity, and decoding these patterns to infer external stimuli is commonly known as brain decoding, has become a central goal in cognitive neuroscience [16, 38, 13]. Functional magnetic resonance imaging (fMRI), due to its non-invasive nature and full-brain coverage, is widely used for studying the neural basis of perception, especially in visual paradigms [39, 40]. Traditional decoding pipelines generally follow two stages: mapping fMRI signals to high-dimensional image features extracted from pretrained visual models, then reconstructing stimuli via generative models [13, 41, 23]. Early work focused on decoding low-level features like orientation, color, or object categories [17, 42], though these approaches were limited by fixed vocabularies. The rise of deep learning particularly CLIP, GANs, and diffusion models enabled prediction of rich semantic embeddings and realistic image reconstructions from brain data [43, 44, 45, 46]. Diffusion models iteratively denoise latent noise using learned priors, where brain-derived representations serve as conditioning inputs to produce coherent reconstructions [46, 44]. However, most models are trained in a subject-specific manner, limiting their generalizability. To address inter-subject variability, studies have proposed anatomical normalization, functional alignment such as hyperalignment [47], and joint models with shared encoders and subject-specific parameters [48]. Contrastive learning has also been applied to align image embeddings with corresponding neural activations across subjects [33, 49], though mostly within single datasets. Cross-dataset decoding remains underexplored, as most work uses fixed acquisition settings, making them prone to domain shifts in protocols, scanners, and stimuli. To our knowledge, no prior work has systematically addressed decoding across entirely different datasets, which is essential for building transferable systems. This challenge is timely, given the success of foundation models that generalize across domains via diverse training data [50, 51]. In this work, we take a first step toward bridging this gap by proposing a unified model for both cross-subject and cross-dataset brain decoding. Our hybrid architecture uses shared components to capture visual-semantic representations, along with subject- and dataset-specific adaptation layers. We apply contrastive learning across datasets to align fMRI signals with shared image embeddings, grounding them in a common semantic space despite variability in subjects and acquisition. This promotes robustness, generalization, and scalability toward universal brain decoding systems.

## C    Datasets

To evaluate our approach for brain decoding across subjects and datasets, we utilized three publicly available fMRI datasets that are widely recognized in the field of visual neuroscience: the Natural Scenes Dataset (NSD) [30], BOLD5000 [31], and Generic Object Decoding (GOD) [32]. These datasets differ in several critical aspects that introduce inter-dataset variability. NSD was collected using a 7T MRI scanner, whereas BOLD5000 and GOD were acquired on 3T scanners. The visual stimuli also vary: NSD includes natural scenes, BOLD5000 comprises a wide range of everyday objects and scenes, and GOD focuses on generic object categories. Additionally, the datasets differ in subject pool sizes and stimulus presentation protocols. This diversity allows us to assess the generalizability of our framework under varied experimental conditions.

**Natural Scenes Dataset (NSD)**    The NSD offers high-resolution fMRI recordings from eight participants who were shown a large set of naturalistic images drawn from the COCO image collection [52]. Each image was displayed for two seconds, followed by a one-second inter-stimulus interval. For our experiments, we selected data from four participants. For each participant the training set included approximately 8,859 images and 24,980 corresponding fMRI trials, while the testing set comprised 982 images and 2,770 trials. To reduce spatial complexity, voxel were selected based on NSD-defined visual region-of-interest (ROI) masks, targeting around 15,000 voxels. Temporal reduction was achieved using pre-estimated beta coefficients derived from a general linear model.

**BOLD5000**    This dataset consists of fMRI data collected while participants viewed a broad range of natural and object-centric images, including samples from COCO and ImageNet. It comprises 4,916 unique stimuli, with 4,803 of them shown once and 113 presented multiple times, yielding a total of 5,254 trials. Data were recorded from four individuals, three of whom completed the entire

set of trials. We adopted a standardized train-test split from Chen et al [25]: all unique-image trials (4,803 samples) were used for training, while trials corresponding to repeated images (113 samples) were used for evaluation.

**Generic Object Decoding (GOD)**   The GOD dataset contains fMRI measurements from five individuals engaged in either image perception or imagery tasks involving objects from the ImageNet database. During the training phase, subjects viewed 1,200 images spanning 150 object categories, with each category represented by eight unique images. The testing phase included 50 distinct object categories, each represented by a single image repeated across 35 trials. Stimuli were shown for nine seconds per trial. Recordings were organized into discrete runs—24 for training and 35 for testing—acquired using a 3T scanner with an EPI sequence (TR = 3000 ms, TE = 30 ms, voxel size = 3 mm³). Data preprocessing included motion correction, detrending, and spatial normalization. For our experiments, we focused on the visual cortex (VC), extracting around 4,500 voxels per subject.

# D   Results

Table 2: Quantitative comparison of image reconstruction on the NSD dataset using structural (low-level) and semantic (high-level) metrics. Results above the horizontal line are from prior cross-subject studies. Below are our results: "NSD" refers to models trained and evaluated on NSD (cross-subject) and "Cross-model-NSD" represents cross-dataset trained models evaluated on NSD. The "(AUG)" variants incorporate data augmentation during training, demonstrating improved performance across both low- and high-level metrics.

| Method | PixCorr ↑ | SSIM ↑ | AlexNet(2) ↑ | AlexNet(5) ↑ | Inception ↑ | CLIP ↑ | EffNet-B ↓ | SwAV ↓ | Top-1 acc ↑ |
|---|---|---|---|---|---|---|---|---|---|
| Mind-Reader [23] | – | – | – | – | 78.2% | – | – | – | – |
| Takagi *et al.* [15] | – | – | 83.0% | 83.0% | 76.0% | 77.0% | – | – | – |
| Gu *et al.* [19] | .150 | .325 | – | – | – | – | .862 | .465 | – |
| Brain-Diffuser [21] | .254 | .356 | 94.2% | 96.2% | 87.2% | 91.5% | .775 | .423 | – |
| MindEye [14] | **.309** | .323 | **94.7%** | **97.8%** | **93.8%** | **94.1%** | .645 | .367 | – |
| Dream [22] | .274 | .328 | 93.9% | 96.7% | 93.4% | 94.1% | .645 | .418 | – |
| NSD | .071 | .328 | 80.5% | 91.2% | 91.7% | 93.0% | .707 | .398 | 55.8% |
| NSD (AUG) | .077 | .346 | 83.6% | 93.0% | **94.1%** | **94.8%** | .675 | .383 | 61.1% |
| Cross-model-NSD | .069 | .326 | 79.7% | 90.8% | 91.5% | 92.9% | .711 | .401 | 55.5% |
| Cross-model-NSD (AUG) | .064 | .337 | 80.3% | 91.5% | 92.2% | 93.5% | .698 | .392 | 57.8% |

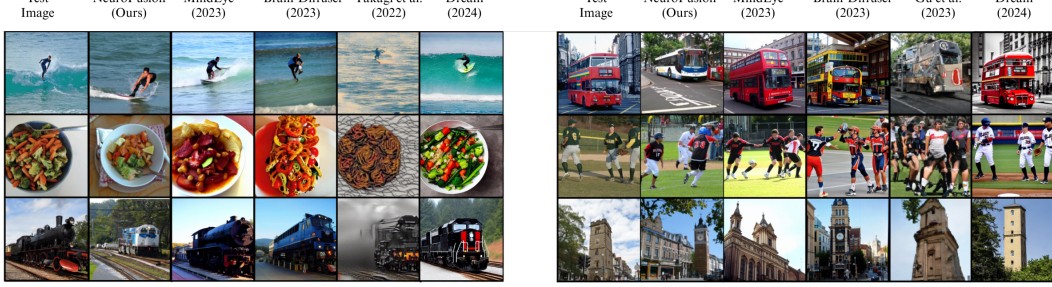

Figure 3: Qualitative reconstructions on NSD. Each row shows an original test image (left) and decoded outputs. Semantic fidelity and structural details are well-preserved across subjects.

Table 3: Quantitative evaluation of image reconstruction on the GOD dataset using structural (low-level) and semantic (high-level) metrics. Results above the horizontal line are from prior cross-subject studies. Below the line are our results: "GOD" denotes cross-subject models trained and evaluated on GOD; "Cross-model-GOD" indicates cross-dataset trained models evaluated on GOD; and "Cross-model-finetuned-GOD" represents those further finetuned on GOD.

| Method | PixCorr ↑ | SSIM ↑ | AlexNet(2) ↑ | AlexNet(5) ↑ | Inception ↑ | CLIP ↑ | EffNet-B ↓ | SwAV ↓ | Top-1 acc ↑ |
|---|---|---|---|---|---|---|---|---|---|
| Koide-Majima [53] | - | - | - | - | - | 90.00% | - | - | - |
| M Ferrante et al [54] | .32 | .38 | 67.00% | 68.00% | 66.00% | 69.00% | - | - | - |
| GOD | .041 | .331 | **79.1%** | **88.5%** | **73.2%** | 84.0% | .884 | .532 | 19.2% |
| Cross-model-GOD | .028 | .337 | **72.3%** | **84.9%** | **68.3%** | 81.0% | .913 | .561 | 16.1% |
| Cross-model-finetuned-GOD | .033 | .313 | **69.1%** | **82.4%** | **68.5%** | 79.8% | .916 | .562 | 15.4% |

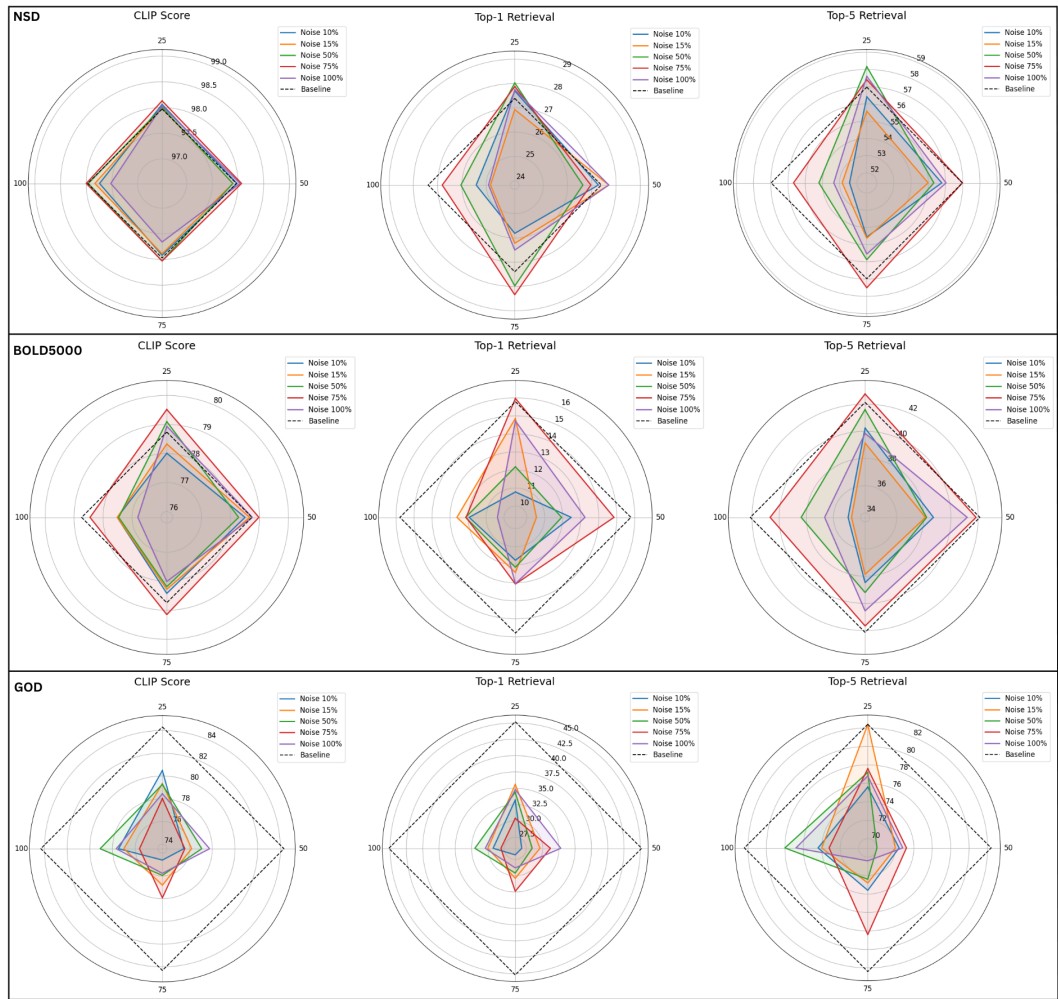

Figure 4: Quantitative evaluation of the data augmentation framework. Each radar plot summarizes the effects of varying augmentation percentages (25%, 50%, 75%, 100%) and noise scales (10, 15, 50, 75, 100) on three key metrics: CLIP Score, Top-1 Retrieval, and Top-5 Retrieval. For each configuration, synthetic fMRI responses were generated by projecting novel image embeddings into subject-specific neural space using ridge regression, followed by controlled noise perturbation sampled from empirical decoding residuals.

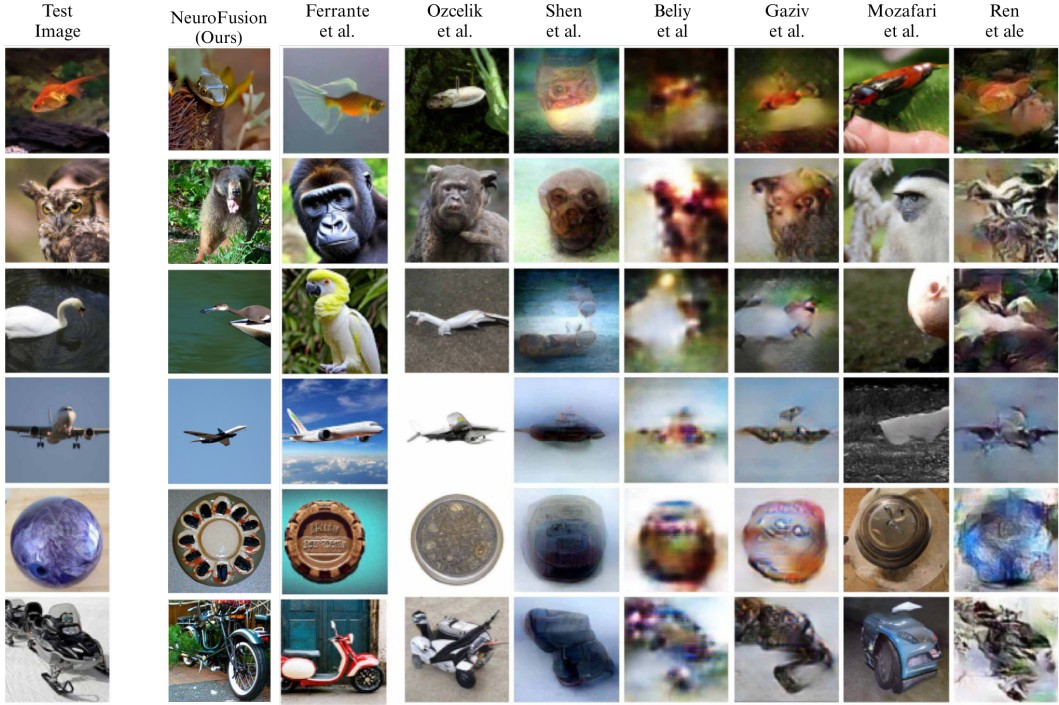

Figure 5: Qualitative comparison of reconstructed images from the GOD dataset. Each row displays a test image (leftmost column) and corresponding reconstructions. The reconstructions are evaluated for both perceptual similarity and semantic fidelity, illustrating the ability of each method to capture visual details and high-level content from brain activity.

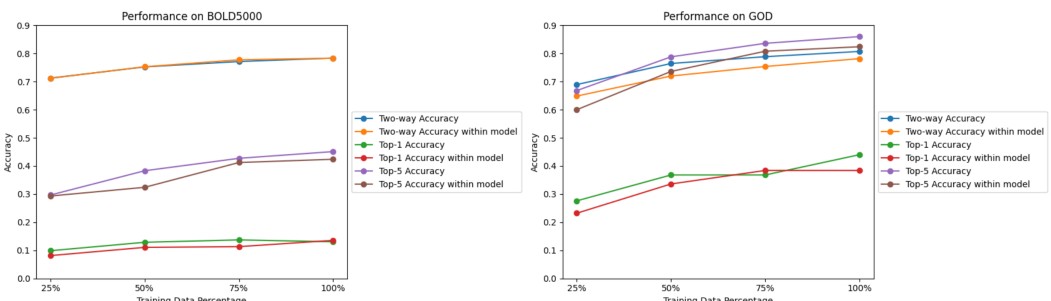

Figure 6: Comparison of model performance on the BOLD5000 (left) and GOD (right) datasets across varying proportions of training data under two training paradigms. In setting I "within-model" approach, models were trained from scratch on each dataset using different amounts of data (cross-subject evaluation). In setting II, a pretrained model (on NSD, BOLD5000, and GOD) was used, with all parameters frozen except the alignment layers, which were finetuned on the target dataset. Results indicate that higher training data percentages lead to improved accuracy, and that finetuning alignment layers enables effective transfer to new datasets with limited data.

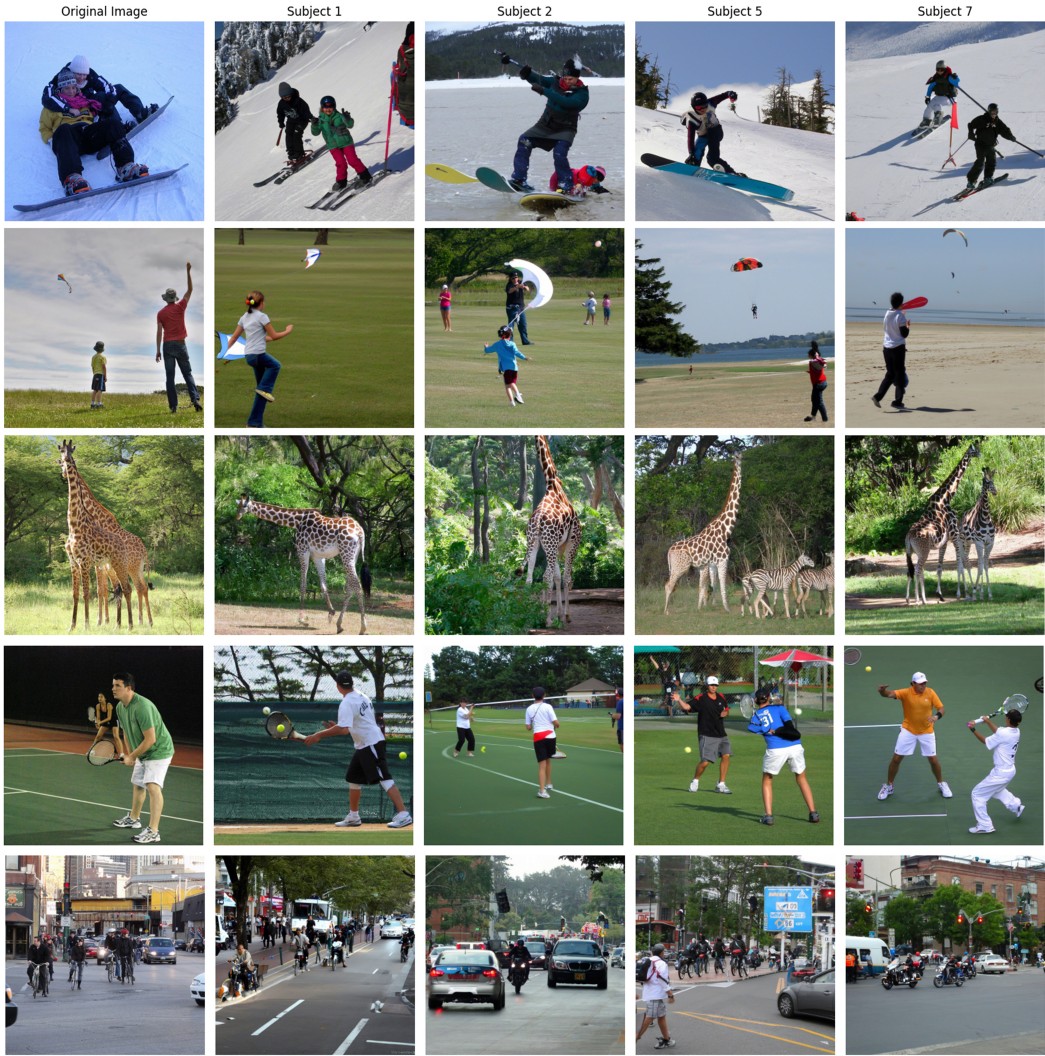

Figure 7: Qualitative reconstruction results NSD dataset. Each row presents a different stimulus image from the test set (leftmost column), with reconstructions generated from the fMRI data of four different subjects (Subjects 1, 2, 5, and 7) shown in the adjacent columns. Despite variations in individual brain signals, the reconstructions preserve the high-level semantic content and overall scene structure across subjects, demonstrating the model's ability to generalize visual decoding.

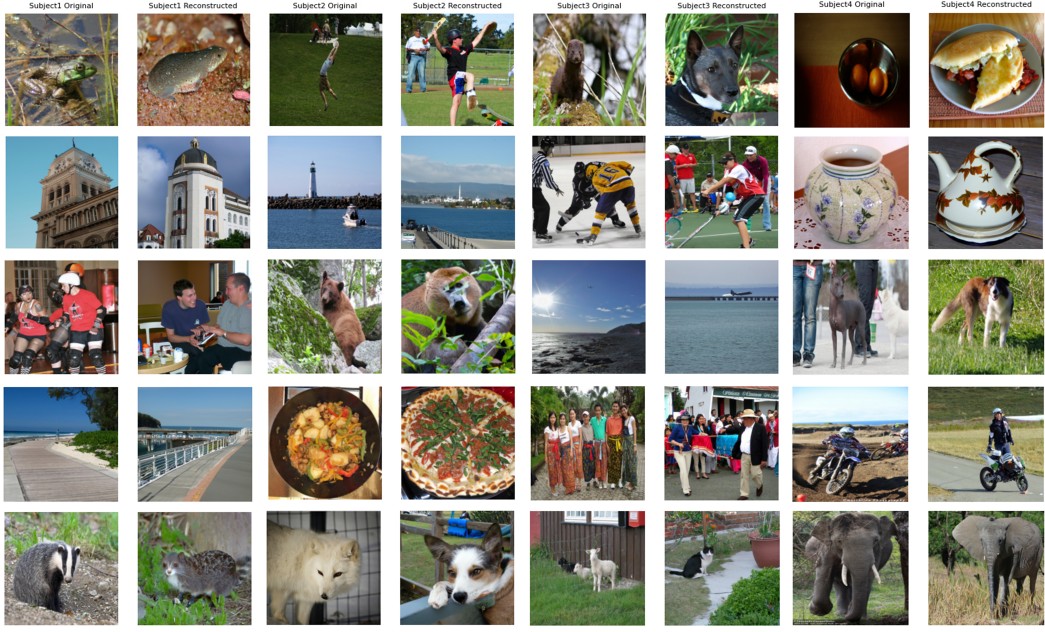

Figure 8: Reconstruction results on the BOLD5000 dataset. Each pair of columns shows original images (left) and the corresponding reconstructions (right) from fMRI signals for four different subjects (Subjects 1–4). The reconstructed images exhibit high-level semantic alignment with their respective stimuli, reflecting the model's ability to decode diverse perceptual content from neural activity.

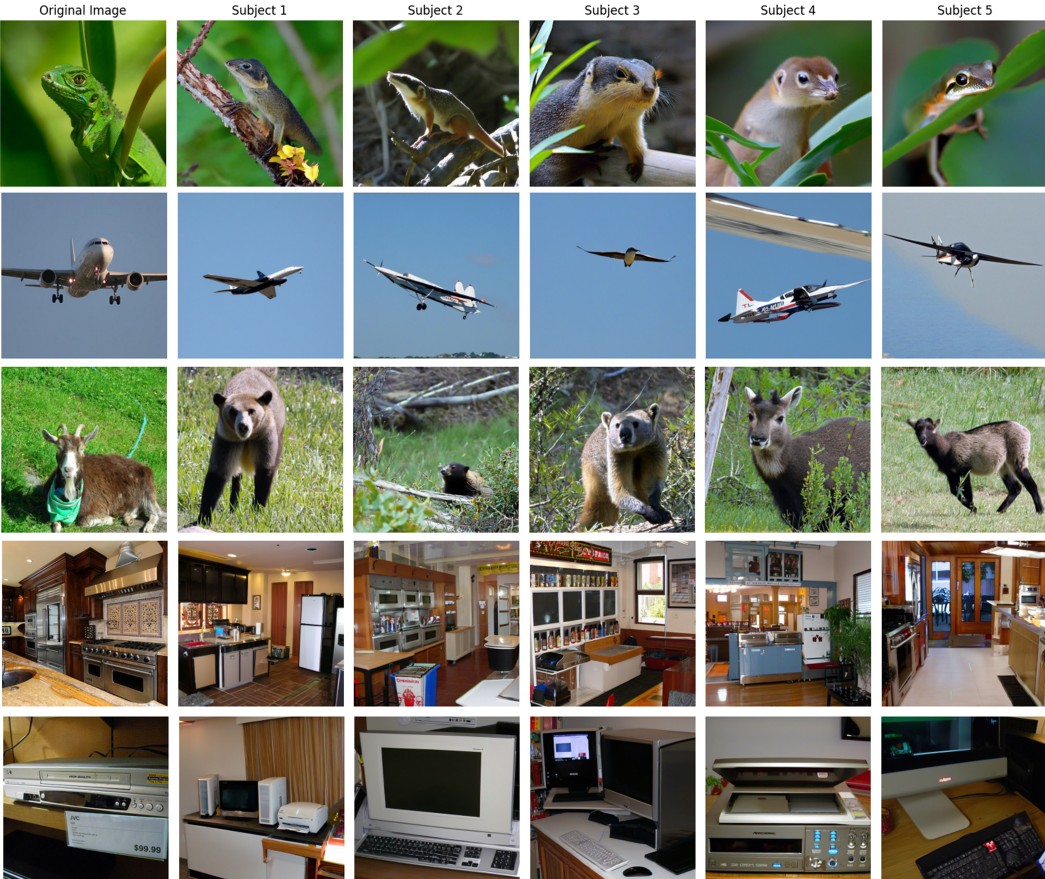

Figure 9: Reconstruction results on the GOD dataset for five different subjects. Each row corresponds to a different stimuli, with the original image shown in the first column and reconstructions from fMRI signals of Subjects 1–5 in the following columns. The reconstructions exhibit strong semantic consistency with the ground truth images.

