# OpenReview forum: "NeuroFusion: A Unified Framework for Generalized Visual Stimulus Decoding from fMRI Across Datasets and Subjects"
_NeurIPS.cc/2025/Workshop/UniReps — UniReps2025_

### Official Review · Reviewer_17KH · 2025-09-14
**Cross-Dataset Brain Decoding: Valuable Empirical Advance with Incremental Methodological Contributions**

**Confidence:** 3

**Review:**

Summary

The paper introduces NeuroFusion, a framework for visual stimulus decoding from fMRI that generalizes across both subjects and datasets. The authors integrate three public fMRI datasets (NSD, BOLD5000, GOD) and employ contrastive learning to align subject-specific neural representations with IP-Adapter/SDXL image embeddings. A ridge regression-based augmentation strategy is used to synthesize fMRI-like signals from novel images. Results show that the framework achieves competitive or state-of-the-art performance on several benchmarks, highlighting the feasibility of scaling decoding across subjects and datasets.

Pros

	•	Timely and important problem: Cross-dataset generalization remains underexplored in brain decoding, and this work takes a clear step toward it.

	•	Strong empirical evidence: Evaluated across three diverse datasets with different scanners, protocols, and subjects; consistently achieves competitive or SOTA results.

	•	Modular and practical framework: Subject-specific alignment layers and simple fine-tuning strategy demonstrate adaptability.

	•	Comprehensive evaluation: Includes qualitative results, ablation studies, and supplementary materials that substantiate claims.

	•	Reproducibility: Code release and clear experimental details improve transparency and credibility.


Cons

	•	Incremental methodology: Core elements (contrastive learning, IP-Adapter, ridge regression) are established; novelty lies in their integration rather than algorithmic innovation.

	•	Simplistic augmentation: Ridge regression with noise sampling is limited in capturing complex neural variability.

	•	Limited analysis of trade-offs: The tension between semantic fidelity (high) and structural fidelity (lower than some baselines) deserves deeper exploration.

	•	Missing statistical rigor: Per-subject variance, error bars, or significance testing are absent despite being common in neuroimaging studies.

	•	Design choices underexplained: Rationale for IP-Adapter over other backbones, or the specifics of subject-specific modules, could be clearer.



Suggestions

	•	Provide ablations using alternative vision-language backbones (e.g., CLIP vs. IP-Adapter).

	•	Include per-subject error bars or statistical analysis where feasible.

	•	Explore augmentation strategies beyond ridge regression (e.g., nonlinear mappings or generative neural priors).

	•	Analyze failure cases to understand when structural vs. semantic fidelity diverges.


Verdict

This paper makes a valuable empirical contribution to brain decoding by tackling cross-dataset generalization, an important and underexplored challenge. While the methodological novelty is incremental, the integration and thorough evaluation across datasets fill a meaningful gap. The results are strong and support broader discussions on universal brain decoders.

**Score:**

4

**Topic Fit:**

2

---

### Official Review · Reviewer_t3dA · 2025-09-14
**Limited Novelty and Weak Neuroscientific Insight**

**Confidence:** 5

**Review:**

The paper presents a popular and important research topic in recent years: reconstructing visual images from brain activity. I have read several recent works that achieve impressive high-resolution image reconstruction using diffusion models and other advanced neural architectures. The authors of this paper also compare their results with some notable methods in the field.

Main Concerns:

1. Lack of Competitive Performance and Novelty

Based on the results presented-specifically in Fig3, where comparisons are made with methods such as MinDEye (2023) and Dream (2024), and in Fig5-it appears that the method by Ferrante et al. still performs significantly better in terms of image similarity compared to orginal images.

Moreover, I am familiar with the MinDEye and Dream papers; their model architectures are not only more advanced but also offer clearly novel contributions. In contrast, this paper does not present a strong case for novelty.

In your abstract, the main claimed contribution appears to be improved generalization "across subjects and datasets," which, while important, is not a new idea and has been explored in prior work.

2. Lack of Neuroscientific Interpretation

The paper does not provide any neuroscientific analysis or interpretation. For example, it would be valuable to understand which brain regions contribute most to the high-resolution reconstructions. Are the improvements due purely to the neural network architecture, or is there an underlying neural mechanism being leveraged?

3. Limited Advancement Beyond Existing Work

If the primary goal of the paper is simply to generate high-resolution reconstructions from brain activity, this alone is no longer a novel contribution. Many recent papers have already demonstrated this capability with superior performance.

While reconstructing visual images from brain signals is undoubtedly a fascinating and important task that continues to attract interest, future work in this area should aim to deepen our understanding of the neural mechanisms behind the reconstructions, not just improve image quality with more powerful models.

Summary:

Overall, I find the contribution of this paper to be limited. The reconstruction quality does not appear to surpass existing methods, and the work lacks both methodological novelty and neuroscientific insight. I encourage the authors to strengthen the novelty of their approach and provide a deeper neuroscientific interpretation to better position this work in the context of the existing literature.

**Score:**

2

**Topic Fit:**

2

---

### Official Review · Reviewer_NmXF · 2025-09-15
**a framework for reconstructing visual stimuli from fMRI signals that generalizes across both subjects and datasets**

**Confidence:** 4

**Review:**

The paper proposes a framework for reconstructing visual stimuli from fMRI signals that generalizes across both subjects and datasets. Unlike prior work that focuses on subject-specific models, NeuroFusion aligns brain activity with vision-language embeddings from IP-Adapter + Stable Diffusion XL (SDXL) using contrastive learning. It also introduces a ridge-regression-based data augmentation method to generate realistic synthetic fMRI signals from novel images, improving robustness and generalization. The model is evaluated on three major datasets (NSD, BOLD5000, GOD)

Advantages:
I would say they are among a few aiming to unify decoding across multiple datasets and subjects instead of training dataset-specific models.
I like that they use IP-Adapter + SDXL embeddings to bypass full retraining, improving efficiency and semantic fidelity.


Comments and concerns:

1- The framework assumes that fMRI signals across different subjects and acquisition protocols can be mapped into a shared semantic embedding space using contrastive learning. However, inter-dataset variability (scanner type, voxel resolution, ROI definitions) might cause non-linear distortions that contrastive alignment + ridge regression cannot fully address.
2- The proposed ridge-regression-based data augmentation assumes a linear mapping between image embeddings and voxel responses. While other generative strategies (e.g., diffusion-based fMRI synthesis) could possibly provide a better realism.
Also, the assumption that a linear model can adequately capture the complex, non-linear relationship between high-level image embeddings and voxel-level brain activity is a significant oversimplification.

3- Although the method achieves high CLIP and Inception similarity scores (semantic metrics), pixel-level reconstruction quality (SSIM, PixCorr) still lags behind models like MindEye. This trade-off between semantic understanding and low-level detail fidelity should be discussed more critically.
4- Despite aiming for universal decoding, the model is trained and tested on only 13 subjects across three datasets, all in the visual domain. It's unclear whether the approach scales to more diverse stimuli or modalities.

**Score:**

3

**Topic Fit:**

2

---

### Official Review · Reviewer_ck3Q · 2025-09-16
**Novel alignment method with potential, but interpretability limited**

**Confidence:** 3

**Review:**

Summary
- Proposes NeuroFusion, a unified framework for visual stimulus decoding from fMRI across both subjects and datasets.
- Uses contrastive learning to align fMRI representations with image embeddings (IP-Adapter + SDXL).
- Adds a simple data augmentation method using ridge regression to synthesize fMRI-like signals with realistic noise.
- Demonstrates competitive / state-of-the-art results across NSD, BOLD5000, and GOD datasets.

Strengths
- Timely and ambitious: cross-dataset decoding is an important step toward more generalizable neural decoding.
- Novel combination of contrastive alignment and ridge-based augmentation, improving robustness.
- Evaluates across multiple large public datasets with clear quantitative and qualitative results.
- Shows both semantic-level decoding (CLIP, Inception similarity) and structural metrics (SSIM, PixCorr).
- Strong potential for workshop discussion on unified decoding and foundation-model approaches to brain decoding.

Weaknesses (beyond those described in the Discussion)
- Reconstructions prioritize semantic fidelity over structural detail; pixel-level resemblance remains weaker than some baselines.
- Comparisons to prior work are thorough but some important baselines (e.g., recent hyperalignment-style or multimodal alignment approaches) could be discussed more.
- No statistical uncertainty measures (error bars, variance across subjects) reported, which makes robustness harder to assess.

Suggestions / Questions
- Could nonlinear or generative augmentation methods (beyond ridge regression) yield more biologically plausible signals?
 - How sensitive is performance to λ in the ridge regression augmentation and noise modeling?
- Could this approach be benchmarked against hyperalignment or shared latent space models explicitly to situate novelty more clearly?
- How well does the model transfer when trained on smaller subsets of datasets (low-data regimes)?

Overall Assessment
- Strong and relevant contribution with clear novelty (first cross-dataset unified decoding framework).
- Results are competitive and sometimes state-of-the-art, though interpretability and structural fidelity remain open challenges.
- Recommendation: Accept (Extended Abstract) – will spark valuable discussion on generalization and neural foundation models.

**Score:**

4

**Topic Fit:**

2